# Dissecting the Molecular Regulation of Natural Variation in Growth and Senescence of Two *Eutrema salsugineum* Ecotypes

**DOI:** 10.3390/ijms23116124

**Published:** 2022-05-30

**Authors:** Fanhua Wang, Zhibin Sun, Min Zhu, Qikun Zhang, Yufei Sun, Wei Sun, Chunxia Wu, Tongtong Li, Yiwu Zhao, Changle Ma, Hui Zhang, Yanxiu Zhao, Zenglan Wang

**Affiliations:** 1Shandong Provincial Key Laboratory of Plant Stress, College of Life Sciences, Shandong Normal University, Jinan 250014, China; wangfanhuacaas@163.com (F.W.); sunzb@ysfri.ac.cn (Z.S.); zhumin020841@163.com (M.Z.); zhangqikun1016@163.com (Q.Z.); iceplant@sdnu.edu.cn (Y.S.); sunwei57@aliyun.com (W.S.); cxwu1001@163.com (C.W.); litongtong020837@163.com (T.L.); zywkaka@163.com (Y.Z.); machangle@sdnu.edu.cn (C.M.); laohanzhang@hotmail.com (H.Z.); 2Yellow Sea Fisheries Research Institute, Chinese Academy of Fishery Sciences, Qingdao 266071, China

**Keywords:** *Eutrema salsugineum*, salt tolerance, natural variation, growth, senescence, molecular regulation

## Abstract

Salt cress (*Eutrema salsugineum*, aka *Thellungiella salsuginea*) is an extremophile and a close relative of *Arabidopsis thaliana*. To understand the mechanism of selection of complex traits under natural variation, we analyzed the physiological and proteomic differences between Shandong (SD) and Xinjiang (XJ) ecotypes. The SD ecotype has dark green leaves, short and flat leaves, and more conspicuous taproots, and the XJ ecotype had greater biomass and showed clear signs of senescence or leaf shedding with age. After 2-DE separation and ESI-MS/MS identification, between 25 and 28 differentially expressed protein spots were identified in shoots and roots, respectively. The proteins identified in shoots are mainly involved in cellular metabolic processes, stress responses, responses to abiotic stimuli, and aging responses, while those identified in roots are mainly involved in small-molecule metabolic processes, oxidation-reduction processes, and responses to abiotic stimuli. Our data revealed the evolutionary differences at the protein level between these two ecotypes. Namely, in the evolution of salt tolerance, the SD ecotype highly expressed some stress-related proteins to structurally adapt to the high salt environment in the Yellow River Delta, whereas the XJ ecotype utilizes the specialized energy metabolism to support this evolution of the short-lived xerophytes in the Xinjiang region.

## 1. Introduction

Salt cress (*Eutrema salsugineum*, also known as *Thellungiella salsuginea* or *Thellungiella halophila* in older literature) is a typical halophyte to study abiotic stress tolerance [1,2,3,4,5]. Not only does it have a high tolerance to extreme salt stress, but it also has a high tolerance to freezing, nitrogen deficiency, and drought stress [2,4,6,7]. From the perspective of genetic research, *E. salsugineum* has the characteristics of small plant size, a short life cycle, a large number of seeds, and strong self-pollination ability. In addition, *E. salsugineum* and *Arabidopsis thaliana* show a high degree of sequence identity at both the cDNA and amino acid levels [8,9,10].

The ubiquitous heterogeneity of natural habitats makes it difficult to extract plant resources and seriously affects the growth, reproduction, and distribution of plants [11,12]. Phenotypic plasticity is an important ecological strategy for plants to adapt to heterogeneous habitats [13]. Plants growing in heterogeneous habitats have high phenotypic plasticity, which can maximize resource acquisition and optimize resource allocation between organs and tissues, thereby improving resource utilization efficiency and ultimately improving plant adaptability. Phenotypic plasticity can play a key role in the successful colonization of adaptive halophytes such as *Eutrema*, and this plasticity can lead to the formation of ecotypes in geographical regions. Ecotypes are common in many plants and are used to describe genetically distinct populations adapted to specific environmental conditions [14,15]. Local selection of different habitats plays a crucial role in the generation and maintenance of genetic diversity [16].

In China, salt cress is distributed from the Yellow River basin to the Yellow River estuary in Dongying City, Shandong Province, resulting in many naturally occurring ecotypes. Among them, Shandong and Xinjiang ecotypes grow in different natural habitats and are typical representatives. Xinjiang is located inland and has a continental temperate dry climate. Dongying City is located in the Yellow River Delta and has a temperate monsoon climate with frequent rainfall [17]. To adapt to the different living environments, the two salt cress ecotypes can adopt complex mechanisms, including developmental, morphological, physiological, and biochemical coping and adaptation strategies.

Current research on salt cress has focused on comparing its response to abiotic stress with that of *A. thaliana* [18]. However, comparative studies on the growth, development, and responses to abiotic stress of different salt cress ecotypes are still sporadic, and comparative studies on the proteomics of different salt cress ecotypes have not yet been reported. As we know, proteome analysis can help us gain more intuitive and accurate knowledge to uncover natural variations. In this study, two *E. salsugineum* ecotypes, Shandong (SD) and Xinjiang (XJ), were selected to study their morphological and physiological differences. In addition, proteomic comparisons were performed under optimal growth conditions to reveal the evolutionary differences at the protein level between these two ecotypes.

## 2. Results

### 2.1. Morphological and Physiological Comparison

After culturing the two *Eutrema* ecotypes in a hydroponic system for 6 weeks, there were significant differences in the morphology of shoots and roots between the two ecotypes. The leaves of the SD ecotype *Eutrema* are dark green, short and flat; The leaves of the XJ ecotype *Eutrema* are light green, long and curly (Figure 1A,B). Although the root system of each ecotype is the taproot, there are major differences between root systems. SD *Eutrema* ecotype has an obvious taproot, and XJ *Eutrema* ecotype has a short taproot and developed lateral roots (Figure 1A). In addition, the SD ecotype *Eutrema* had longer taproots than the XJ ecotype (Figure 1D). Biomass analysis showed that the fresh weight of shoots of XJ *Eutrema* was higher, while the difference in fresh weight of roots was not significant (Figure 1E). The average dry weight of the XJ ecotype was also larger than that of the SD ecotype (Figure 1F). When comparing the leaf inclination angle of the two ecotypes, we found that the leaf pitch angle of the SD ecotype *Eutrema* was significantly greater than that of the XJ ecotype (Figure 1C,G). The IAA concentration of the SD ecotype was significantly higher than that of the XJ ecotype, which may be closely related to leaf margin shape. (Figure 1C,H) [19].

During hydroponics, we found that the XJ ecotype *Eutrema* showed clear signs of senescence or leaf loss (Figure 2A). Therefore, we compared the degree of leaf senescence of the two ecotypes and found that the number of etiolated leaves and the chlorosis extent of same-position leaves of the XJ ecotype were higher than those of the SD ecotype, showing obvious leaf yellowing symptoms (Figure 2B). Meanwhile, the chlorophyll content of the SD ecotype leaves was about 1.7 times higher than that of the XJ ecotype (Figure 2C). This result is consistent with the phenomenon of leaf senescence. Further comparing the leaf area of the two *Eutrema*, we found that the leaf area of the XJ ecotype was significantly larger than that of the SD ecotype (Figure 2D). To elucidate the role of phytohormones in *Eutrema* leaf senescence, the cytokinin (CK) concentrations of the SD ecotype were measured, and the results showed that the SD ecotype had significantly higher cytokinin concentrations than the XJ ecotype. (Figure 2E). This result suggests that cytokinin (CK) plays an important role in leaf senescence in both ecotypes. To further elucidate the mechanism of the leaf senescence differences between the two ecotypes, the expression of leaf senescence marker genes senescence-associated gene 12 (*SAG12*) and *SAG113* was analyzed. Additionally, we found that the expression of these two genes was decreased in the SD ecotype (Figure 2F), which reflected the molecular basis of the difference in leaf senescence between these two ecotypes. Therefore, our results suggest that the XJ ecotype leaves are more susceptible to age-dependent senescence.

### 2.2. Identification of Differently Expressed Protein Spots from Two Ecotypes by 2-DE and ESI-MS/MS

To investigate the mechanisms underlying the morphological and physiological differences between the two *Eutrema* ecotypes, a comparative proteomic analysis of *Eutrema* was performed. Proteins were extracted from the shoots or roots of 6-week-old seedlings of both ecotypes and separated by IEF/SDS-PAGE, respectively. After image analysis, more than 900 protein spots were reproducibly detected and matched in the shoot gel, and more than 700 protein spots were reproducibly detected and matched in the root gel. A representative 2-DE gel image is shown in Figure 3. Proteins were well separated in both dimensions. The isoelectric points (pI) of the spots range from 4 to 7, and the molecular mass ranges from 10 to 80 kDa. Only the protein spots exhibiting significant changes (>2 fold or <0.5 fold and *p*-value < 0.05) between different ecotypes were employed for further analysis.

Quantitative image analysis revealed that 66 protein spots from shoot gels and 34 protein spots from root gels showed significant differences between the two ecotype samples (Figure 3). Most proteins in shoots were completely absent in SD or XJ ecotype samples, and some protein spots were increased or decreased in SD or XJ ecotype samples. While most proteins in roots were characterized by the acid-base properties of pI, only 4 proteins appeared in SD or XJ ecotype samples compared with each other. The differentially expressed protein spots were excised, digested with trypsin, and identified by ESI-MS/MS. The identified protein has two or more peptide fragments (Appendix A). The results showed that 25 protein spots in shoots and 28 protein spots in roots were identified by ESI-MS/MS (Table 1 and Table 2).

Interestingly, three proteins were identified in shoots in the *Eutrema* SD ecotype, such as 2,3-bisphosphoglycerate-independent phosphoglycerate mutase 1 (spots SS4 and 5), plasma membrane-associated cation-binding protein 1 (spots SS6, 7, and 8), and 2-Cys peroxiredoxin BAS1 (spots SS11, 12). In the roots of the SD ecotype *Eutrema*, jacalin-related lectin 34 (spots SR1, 2, and 3) and 2,3-bisphosphoglycerate-independent phosphoglycerate mutase 1 (spots SR5, 6, and 7) were identified. Likewise, the 2,3-bisphosphoglycerate-independent phosphoglycerate mutase 1 was found in the roots of the XJ ecotype *Eutrema* (spot XR2, 3, and 7). Further examination of the gel map shows that the experimental values of isoelectric point or molecular mass deviate from the theoretical values. This phenomenon may be due to the presence of different protein isoforms and post-translational modifications or degradation, which may alter the protein’s molecular weight and/or charge. Alternatively, proteins appearing at multiple sites may be due to the translation of alternatively spliced mRNAs [20].

### 2.3. Functional Annotation of Different Proteins

To further functionally classify differentially expressed proteins, GO enrichment analysis following biological processes was performed. Representative GO terms were involved in the cellular metabolic process (24%), response to stress (18%), response to abiotic stimulus (10%), and aging (4%), etc., in the two ecotype shoots (Figure 4A). In the root system, the stress response was the largest (18.6%), followed by the small molecule metabolic process (16.28%), oxidation-reduction process (13.95%), and the response to abiotic stimulus (13.95%) (Figure 4B). This finding suggests that metabolism-related proteins and stress-related and defense-related proteins might play important roles in the evolution of distinct intrinsic traits of the two ecotypes.

### 2.4. Analysis of Differentially Expressed Protein Spots from Two Ecotypes

Proteins associated with cellular metabolic processes such as ketoacid reductase isomerase (KARI) (spot XS4, Figure 5A), glutathione synthetase (GSH2) (spot XS19, Figure 5A), and L-diaminopimelate aminotransferase (AGD2) (spot XS8, Table 1) appear in shoots of the XJ ecotype. As a major bifunctional enzyme, KARI catalyzes a two-step reaction in branched-chain amino acid biosynthesis to produce the precursors of valine, leucine, and isoleucine [21]. Glutathione synthetase encodes the enzyme that converts γ-glutamylcysteine (γ-EC) to glutathione. In plants, glutathione, as a powerful non-enzymatic antioxidant, plays a key role in various physiological responses such as redox homeostasis and ROS scavenging, detoxification of heavy metals, and development [22,23,24]. Under normal conditions, root growth is reduced due to impaired glutathione biosynthesis [25]. LL-diaminopimelate aminotransferase plays a role in plant lysine biosynthesis [26]. Based on these results, we believe that the increase in metabolism-related proteins helps XJ ecotype plants generate more energy from carbon and nitrogen assimilation, leading to faster growth and increased biomass, which is consistent with our previous physiological evidence. On the other hand, the contents of other proteins related to cellular metabolic processes were also increased in SD ecotype plants. Among them, 2,3-bisphosphoglycerate-independent phosphoglycerate mutase 1 (iPGAM1) (spots SS4 and 5, Figure 5A) catalyzes the reversible conversion of 3-phosphoglycerate to 2-phosphoglycerate during glycolysis. ATP sulfurylase 1 (APS1) (spots SS27, Figure 5A) increased less than 2-fold in SD ecotype plants shoot compared to XJ ecotype. APS catalyzes the first step of sulfate assimilation in plant plastids and cytosol, activating inorganic sulfate to adenosine-5′-phosphosulfate, which, in turn, is converted to a variety of sulfides such as cysteine (Cys), methionine (Met), glutathione (GSH) associated with plant tolerance to various abiotic stresses [27]. Therefore, we hypothesized that increasing APS1 in SD ecotypes would accelerate sulfur assimilation and improve SD ecotype tolerance to abiotic and biotic stresses.

In addition, four proteins involved in the stress response were significantly elevated in the SD ecotype, including plasma membrane-associated cation-binding protein 1 (PCAP1) (spots SS6, 7, and 8, Figure 5B), pathogenicity-associated protein 5 (PR5) (spot SS10, Figure 5B), endochitinase (CHI) (spot SS14, Figure 5B) and glucan endo-1,3-beta-glucosidase (BG1) (spot SS29, Figure 5B). PCAP1 is a hydrophilic cation-binding protein that localizes to the plasma membrane via N-myristoylation on glycine 2 and interacts with calmodulin and phosphatidylinositol phosphate [28]. PR5, CHI, and BG1 are pathogenesis-related (PR) proteins that are key components of the plant’s innate immune system, particularly systemic acquired resistance (SAR) [29]. In addition to biotic stress, the PR gene can also be induced by a variety of abiotic stresses such as salt, drought, and cold, which enhances the resistance against abiotic stress [30]. In this study, the expression of three PR proteins, PR5, CHI, and BG1, were upregulated in SD ecotype plants, suggesting that SD ecotype plants may have a higher tolerance to biotic and abiotic stress than the XJ ecotype. In addition to the SD ecotype, the XJ ecotype shoots also show a small increase in abiotic stimulation-related proteins such as 20 kDa chaperonin (Cpn20) (spot XS12, Figure 5B). Chloroplast Cpn20 is a plastid-specific co-chaperone that is essential for assisting Cpn60 in protein folding. Meanwhile, Cpn20 might be an iron chaperone for iron superoxide dismutase (FeSOD) activation, independent of its co-chaperonin role in the *Arabidopsis* chloroplasts [31,32]. Therefore, we speculate that the high expression of Cpn20 in leaves may confer resistance to the abiotic environment in XJ ecotype plants, thereby promoting the growth and accumulation of above-ground biomass in XJ ecotype plants.

1-Aminocyclopropane-1-carboxylic acid oxidase (ACO), an aging-related protein, is the rate-limiting enzyme for ethylene production in certain dedicated processes [33]. Ethylene affects plant growth and development, including fruit ripening and leaf senescence [34]. In this report, we found that the expression of 1-aminocyclopropane-1-carboxylate oxidase 4 (ACO4) (spot XS6, Figure 5C) in shoots of the XJ ecotype was significantly higher than that of the SD ecotype. Further analysis of ACO concentrations in both ecotypes also showed a trend consistent with ACO4 expression (Figure 5D). These results suggest that upregulation of ACO4 protein expression in leaves of the XJ ecotype may lead to earlier leaf senescence and a shorter life cycle compared with the SD ecotype.

In the roots proteome, various proteins have been detected during the small molecule metabolic process, such as 2,3-bisphosphoglycerate-independent phosphoglycerate mutase 1 (iPGAM1) (spots SR5, 6, and 7, spots XR7, 2, and 3, Figure 6A), pyruvate kinase (PK) (spot XR5, Table 2) and pyruvate dehydrogenase E1 component subunit beta-1 (MAB1) (spots SR16, XR9, Figure 6B). In our study, the differences in iPGAM1 were mainly reflected in isoelectric point shifts between SD and XJ ecotypes. Phosphoglycerate mutase catalyzes the interconversion of 3-phosphoglycerate to 2-phosphoglycerate [35]. This glycolytic enzyme has been reported to be a key component in providing energy and/or metabolites for multiple metabolic pathways [36]. Pyruvate kinase (PK) is a key metabolic enzyme that catalyzes the final step of glycolysis, transferring a high-energy phosphate group from phosphoenolpyruvate (PEP) to ADP to generate ATP and pyruvate [37]. In our study, PK was present in the roots of XJ *Eutrema* but not in the roots of the SD ecotype. Likewise, MAB1 also increased in the roots of the XJ ecotype. In plants, the mitochondrial pyruvate dehydrogenase complex consists of the following three structural components: E1, E2, and E3. E1 (pyruvate dehydrogenase) is responsible for the oxidative decarboxylation of pyruvate. Based on the above results, we speculate that the increase in root metabolic proteins in the XJ ecotype may contribute to the uptake of sufficient inorganic elements to maintain a higher aboveground biomass than the SD ecotype.

In addition to metabolism-related proteins, oxidation-reduction process-related proteins were increased or present only in XJ ecotype roots, including NADH dehydrogenase [ubiquinone] iron-sulfur protein 1 (CI76) (spot XR1, Table 2), NAD(P)H dehydrogenase (quinone) (FQR1) (spot XR13, Table 2), monodehydroascorbate reductase 2 (MDAR2) (spots SR8, XR8, Figure 6C). In particular, methylmalonate-semialdehyde dehydrogenase (MMSDH) (spot XR6, Table 2) was highly expressed in XJ *Eutrema* roots but not in the SD ecotypes. CI76 encodes the subunit of the 400 kDa subcomplex of the mitochondrial NADH dehydrogenase (complex I), the first complex of the respiratory chain and the main entrance site for electrons into the respiratory electron transfer chain, and plays a role in maintaining redox balance in plant cells [38]. FQR1 belongs to the family of flavin mononucleotide-binding quinone reductases, catalyzes the electron transfer of NADH and NADPH to multiple substrates, and functions as a quinone reductase in plants. Its gene, *FQR1,* is the main gene of the auxin response. Hence, it is speculated that FQR1 might be involved as a detoxification enzyme in the auxin-induced redox process [39]. As a key component of the ascorbate-glutathione cycle, MDAR2 plays a role in scavenging toxic reactive oxygen species, such as H_2_O_2_, a byproduct of aerobic metabolism in plant chloroplasts, mitochondria, and peroxisomes [40]. Aldehyde dehydrogenases (ALDHs) represent a protein superfamily of NAD(P)^+^-dependent enzymes that oxidize various endogenous and exogenous aliphatic and aromatic aldehydes to the corresponding carboxylic acids. ALDHs play an important role in regulating aldehyde homeostasis, and overexpression of some ALDHs can improve abiotic stress tolerance in plants [41]. Different expression patterns of *ALDH7B4* and *ALDH10A8* in Arabidopsis and *E. salsugineum* contribute to salt tolerance [42]. MMSDH (ALDH6B2), a member of the ALDH family, plays an important role in root development and leaf sheath elongation in rice [43]. Collectively, these antioxidant proteins play an important role in maintaining root redox homeostasis in the XJ ecotype and providing protection for root adaptation to the environment.

Some proteins that respond to abiotic stimuli are also found in the root proteome, such as alcohol dehydrogenase class-P (ADH1) (spot SR9, Table 2), glutathione S-transferase F10 (GST PHI10) (spots SR10, XR10, Figure 6D), 20 kDa chaperonin (Cpn20) (spots SR11, XR11, Figure 6D), paraxanthine methyltransferase 1 (PXMT1) (spot SR15, Figure 6E) and jacalin-related lectin 34 (JRL34) (spots SR1, 2, 3 and spot XR4, Figure 6F). The protein expression levels of ADH1 and PXMT1 in the SD ecotype were higher than those in the XJ ecotype, while GST PHI10 and Cpn20 were up-regulated in the roots of the XJ ecotype. ADH1 catalyzes the reversible conversion of acetaldehyde to ethanol while simultaneously oxidizing NADH to NAD^+^ in response to hypoxic stress. This is essential for Arabidopsis survival under hypoxic conditions and contributes to other biotic and abiotic stress reactions [44]. Plant glutathione S-transferases (GSTs) are a class of multifunctional proteins that are induced by a variety of stimuli. GST PHI10 is a member of the Phi subfamily, particularly in plants [45]. Furthermore, we found in our study that the isoelectric point of JRL34 is significantly different between the two *Eutrema* ecotypes. Studies have shown that this protein is a phosphorylated protein [46], so we speculate that post-translational modifications may contribute to the isoelectric point difference of this protein between the two ecotypes. In addition, jacalin-related lectins (JRLs) are a subset of proteins binding carbohydrates and having one or more jacalin domains. Many JRLs have been shown to be associated with resistance to abiotic and biotic stresses and are induced by stress hormones such as ABA, SA, and JA [47]. Therefore, we speculate that changes in JRL34 may partly determine the differences in the responses of the two ecotypes to different abiotic stimuli.

### 2.5. Comparison the Expression Patterns of Genes Encoding Some Differential Proteins

To examine changes in gene expression at the mRNA level, qRT-PCR analysis was performed on randomly selected genes encoding some differently expressed proteins in 6-week-old seedlings under hydroponic conditions.

Compared with the XJ ecotype, we found that under normal conditions, the SD ecotype had increased the expression of two genes in plants, namely, PR5 and CHI. In contrast, five genes, *KARI*, *ACO4*, *AGD2*, *GSH2,* and *BG1*, were significantly up-regulated in XJ ecotype plants. Among them, the abundance of *ACO4* transcripts in XJ ecotype shoots was significantly higher than that in SD shoots, which was consistent with the expression of ACO4 at the protein level (Figure 7A).

Changes in genes encoding root differential proteins were compared at the mRNA level. As shown in Figure 7B, genes and proteins such as *ADH1*, *PXMT1*, *MMSDH*, *GST PHI10,* and *Cpn 20* had similar patterns of alteration. However, some root DEPs showed inconsistent expression patterns at the mRNA and protein levels. Our results confirm that gene expression at the transcriptional level does not correlate well with expression at the protein level [48], underscoring the importance of using the proteome to reveal the biochemical mechanisms of natural variation between the two *Eutrema* ecotypes.

## 3. Discussion

Salt cress is an extremophile that has been proposed as a model for studying the mechanism of abiotic stress tolerance and is widespread throughout the world [1]. The broad geographic distribution encompasses substantial variation in growth environments, and phenotypic variation among accessions is expected to reflect the genetic variation that is important for adaptation to specific conditions. In recent years, research based on the analysis of the natural genetic variation of species has received increasing attention [49,50]. Until recently, the physiological mechanisms adopted by two ecotypes of SD and XJ *Eutrema* to perceive and acclimate to their environment were seldom discussed in a biochemical context or from a biochemical perspective. In this study, a proteomic approach was used to compare the morpho-physiological properties of two different *Eutrema* ecotypes and their protein expression profiles in shoots and roots. Through the comprehensive analysis of these differentially expressed proteins, we have a preliminary understanding of the protein-level regulatory mechanism behind the phenotypic differences between the two ecotypes and provide important information for understanding the natural mechanism of variation of *Eutrema*.

### 3.1. Variations in Morpho-Physiological Traits

The two ecotypes of salt cress seedlings had great differences in morphological and physiological characteristics. The biomass and leaf area of XJ ecotype plants are larger than those of SD ecotype plants. While the primary root system of SD ecotype plants is more significant than that of XJ ecotype plants, the leaf inclination angle is larger, and the serrated leaf margin is even more evident (Figure 1). These differences may be due to long-term adaptation to different natural habitats, resulting in intrinsic differences in gene expression, metabolic pathways, and hormone levels between the two ecotypes. Furthermore, we found that the expression and concentration of 1-aminocyclopropane-1-carboxylic acid oxidase 4, a key enzyme in ethylene synthesis, was increased in the XJ ecotype (Figure 1C,G and Figure 5C,D), which could be related to leaf senescence and leaf inclination. The previous study indicates that higher levels of ethylene stimulate a more vertical orientation of the petioles (hyponasty) and enhance elongation [51]. There is evidence that differences between species and ecotypes in the effects of ethylene on growth may be related to the altitude of the original habitat [51]. Furthermore, our data suggest that the SD ecotype plants have higher IAA concentrations than the XJ ecotype (Figure 1H) and that intrinsic differences in auxin levels may be reflected in differences in leaf margin shape to a certain extent (Figure 1C) [19]. These changes suggest that morphological and physiological changes between SD and XJ ecotypes are triggered by different regulatory mechanisms.

### 3.2. Stress and Defense-Related Proteins in Shoots

Biological or abiotic stresses reduce crop yields worldwide. Many attempts have been made to confer pathogen resistance and increase abiotic stress tolerance on agronomically valuable plants. Defense-related proteins have been used to alter plant resistance to pathogens and other environmental challenges. In our study, some stress- and defense-related proteins, such as PCAP1 (spots SS6, 7, and 8, Figure 5B), PR5 (spot SS10, Figure 5B), CHI (spot SS14, Figure 5B), and BG1 (spot SS29, Figure 5B), were highly expressed in SD ecotype plants, which may improve plant resistance to various pathogens and abiotic stresses. PCAP1 is a hydrophilic cation-binding protein with the ability to bind Ca^2+^, Mg^2+^, and Cu^2+^. At the same time, it can be fixed to the plasma membrane by N-myristoylation [28,52,53], and participate in intracellular signal transduction by interacting with PtdInsPs and calmodulin [54]. Thus, PCaP1 has multiple physiological roles, including partial involvement in stoma closure [55] and inhibition of microtubule polymerization by binding to tubulin [56]. The occurrence of PCAP1 in SD ecotype *Eutrema* shoots suggests that SD ecotype *Eutrema* may have increased stress tolerance. PR5 is an important defense-related protein in plants and is involved in various stress responses [30]. Chitinases are part of the plant defense system, are nontoxic to plants and higher vertebrates, and are involved in plant defenses against pathogens [57,58]. Previous reports have shown that the expression of chitinase genes increases resistance to various fungal diseases [59,60,61]. Recent studies have shown that chitinase is also involved in the abiotic stress responses of plants, helping plants survive in stressful environments [62]. Beta-1,3-glucanase, also known as pathogenesis-related protein (PR) and found in many plant tissues, catalyzes the hydrolysis of beta-1,3-glucan. β-1,3-glucanase has been shown to participate in defenses against fungal pathogens and abiotic stresses such as salt and drought stress [63,64,65]. In conclusion, compared with the XJ ecotype, the SD ecotype of *Eutrema* strongly expresses some key components of the plant’s innate immune system, especially systemic acquired resistance, thereby improving the plant’s resistance to biotic and abiotic stresses. However, this plant’s fight against stress upsets the balance between energy production and energy expenditure and reduces the plant’s ability to grow [66]. The SD ecotype salt cress, naturally situated in the Yellow River Delta, where most of the soils are salinized to different degrees, may need to adopt this strategy to form and maintain its intrinsic adaptive mechanism to the environment. Therefore, even under normal growth conditions, the SD ecotype salt cress still highly expresses resistance-related proteins that require more metabolites and energy, thereby slowing down its growth. A balanced mechanism between plant growth and stress resistance helps plants develop, maintain growth, and yield in stressful environments.

### 3.3. The Mechanism of Natural Variation of Leaf Senescence between Two Eutrema Ecotypes

Senescence is the final step in leaf development and is usually accompanied by a color change from green to yellow or brown [67]. Leaf yellowing is not only related to age, but can also be caused by many other factors, including biotic stress, mechanical damage, harvesting, darkness, nutrient deficiencies, environmental stress, and phytohormones. The results showed that the rosettes of XJ ecotype *Eutrema* appeared more marked by senescence with aging than those of the SD ecotype (Figure 2A,B). Furthermore, our previous work revealed that the protein expression level and concentration of 1-aminocyclopropane-1-carboxylic acid oxidase 4 (ACO4) were higher in XJ ecotype shoots than in SD ecotype plants (Figure 5C,D). ACO, a key enzyme in ethylene biosynthesis, catalyzes the conversion of 1-aminocyclopropane-1-carboxylic acid to ethylene [68,69], suggesting that XJ ecotype plants have higher ethylene content than SD *Eutrema*. Ethylene is an important gaseous phytohormone that promotes fruit ripening and leaf senescence [34]. Ethylene can only induce leaf senescence from a certain age but cannot directly regulate the onset of leaf senescence [70,71]. Furthermore, glutathione can induce ethylene biosynthesis by regulating the transcription and protein levels of its key enzymes ACS2, ACS6, and ACO1 [72]. Here, we found that GSH2 (spot XS19, Figure 5A) is also present in the XJ ecotype but not in the SD ecotype. As with ACO1, the appearance of ACO4 may be related to glutathione synthetase, leading to increased ethylene synthesis and earlier leaf senescence in XJ ecotype leaves. In a sense, the differences in ethylene content caused by ACO4 and/or GSH2 between the two ecotypes *Eutrema* are the result of their developmental diversity, and conversely, differences in ethylene content boost their distinctions in the leaf senescence process. In addition to ethylene, other plant hormones such as cytokinins and auxins also affect leaf senescence. However, unlike ethylene, both cytokinin and auxin can delay leaf senescence [73,74]. The results also showed that the levels of cytokinin and auxin (IAA) in the XJ ecotype leaves were significantly lower than those of the SD ecotype leaves, consistent with their different leaf senescence symptoms (Figure 1H and Figure 2E). In addition, studies have shown that 2,3-bisphosphoglycerate-independent enzyme 1 (iPGAM1) is also involved in chlorophyll synthesis, photosynthesis, and chloroplast development. The deficiency of a 2,3-bisphosphoglycerate-independent enzyme leads to chlorosis, chloroplast deformities, and impaired photosynthesis [75]. Our result that iPGAM1 is only present in shoots of the SD ecotype (spots SS4 and 5, Figure 5A) can interpret the physiological differences between the two ecotypes, namely, the SD ecotype plants have higher chlorophyll content and a slower rate of aging than the XJ ecotype.

Leaf senescence is also genetically controlled and requires differential expression of specific genes. Among them, a large number of age-related genes, *SAG12*, *SAG13*, and *SAG113,* are upregulated during aging. *SAG12* encodes a cysteine protease and is an important aging-related reference gene, and its encoding protein or mRNA level is significantly increased in aging tissues [76]. *SAG113* encodes a member of the Golgi protein phosphatase 2C family involved in chlorophyll degradation, abscisic acid (ABA) regulation of stomatal motility, and water loss during leaf senescence [77]. Therefore, *SAG12* and *SAG113* play a central role in the aging process. In particular, transcripts of *SAG12* and *SAG113* were also significantly increased in XJ *Eutrema* leaves compared to SD ecotypes (Figure 2F). These results explain the molecular mechanism underlying the different rates of leaf senescence between the two *Eutrema* ecotypes. Combined with previous research and our experimental data, we speculate that ethylene, cytokinin, and auxin signaling may connect other genetic regulators to form a complex regulatory network that regulates the natural variation in leaf senescence in the two *Eutrema* ecotypes.

### 3.4. Proteins Related to Energy Metabolism in Roots

Plant roots need water and nutrients to grow through the soil, which is an important resource for plant growth and productivity [78]. Root growth requires substantial amounts of energy, and this energy comes mainly from the carbohydrates’ catabolism. In addition, aerobic oxidation of carbohydrates is one of the major catabolic pathways, including glycolysis, oxidation of pyruvate to acetyl-CoA, and the tricarboxylic acid (TCA) cycle and subsequent oxidative phosphorylation at the inner membrane of mitochondria [36,79].

Among the many enzymes involved in catabolic processes, 2,3-biphosphoglycerate-independent phosphoglycerate mutase (iPGAM1) is widespread in plants, algae, many invertebrates, fungi, and bacteria, and is relatively conserved among these species, suggesting it plays a key role in maintaining normal glycolysis [80,81]. In glycolysis, iPGAM1 catalyzes the reversible conversion of 3-phosphoglycerate to 2-phosphoglycerate, a precursor generating phosphoenolpyruvate (PEP), a high-energy compound [36]. The difference in iPGAM1 isoelectric points between the two ecotypes may indicate some intrinsic differences in the amino acid sequence or modification state of the two enzymes. Pyruvate kinase (PK) is one of the three rate-limiting enzymes in glycolysis, which catalyzes the transfer of phosphoenolpyruvate to ADP, yielding one molecule of pyruvate and one molecule of ATP [37]. Pyruvate enters the mitochondria to generate ATP via the tricarboxylic acid (TCA) cycle and oxidative phosphorylation [82]. Increasing evidence suggests that PK may play an important role in cell growth [83]. In plants, PK is an enzyme present in cytoplasmic (PKc) and plastid (PKp) isozymes [84]. Downregulation of PKc in T-DNA insertion mutants affects the glycolytic pathway, leading to dwarfism [85]. Therefore, the occurrence of PK only in the roots of XJ plants is of great importance for growth and development. The mitochondrial pyruvate dehydrogenase complex consists of the following three components: E1, E2, and E3 in all organisms and is the main entry point for carbon into the tricarboxylic acid cycle. The E1 component of the pyruvate dehydrogenase complex consists of an E1α catalytic subunit and an E1β regulatory subunit. In the experimental results, we found that one of the important components of this complex, the pyruvate dehydrogenase E1 component subunit β-1 (MAB1), was more strongly expressed in XJ roots than in SD roots, indicating that MAB1 has an important role in the XJ ecotype root metabolism.

Glycolysis and the tricarboxylic acid (TCA) cycle are two important pathways for the aerobic oxidation of carbohydrates, which are responsible for providing energy and carbon skeletons, and are essential for various physiological activities and morphogenesis in plants. The above-mentioned proteins involved in the aerobic oxidation of carbohydrates play an important role in both energy production and plant growth and development. Therefore, the accumulation of these proteins in the roots of the XJ ecotype can help roots absorb enough inorganic elements from the external environment to support the growth of the aerial parts.

## 4. Materials and Methods

### 4.1. Plant Materials and Growth Conditions

About 100 seeds were surface sterilized using a sodium hypochlorite solution (0.7% available chlorine) containing 0.1% (*v*/*v*) Tween 80 for 10 min. The seeds were washed 6 times, then spread on solid MS medium (MS + 3% sucrose (*w*/*v*) + 0.7% (*w*/*v*) agar, pH 5.8) and stratified in the dark at 4 °C for 7 days. After stratification, MS plates were transferred into a growth chamber (20 ± 2 °C, 75% relative humidity, 16 h/8 h light-dark regime, under an optimal light intensity of 110 μmol m^−2^ s^−1^). Four-day-old seedlings were transferred from MS plates onto the hydroponic culture system (the detail see Appendix A).

### 4.2. Growth Parameter Measurements and Morphological Analysis

After 6 weeks of growth in the hydroponic system, the primary root lengths of SD group and XJ group were measured, respectively. The fresh weights (FW) of shoots and roots were determined immediately after sampling. The dry weight (DW) was then determined after drying the shoots and roots at 80 °C for 48 h. Nine plants were used for each biological replicate and 3 independent biological replicates were produced for each ecotype. To analyze the morphological characteristics of each ecotype, we measured leaf inclination and leaf area. After the hydroponic seedlings were grown for 6 weeks, the shoots of SD and XJ plants were taken out and cut into two longitudinally with the central axis of the seedling as the center. The angle between the central axis and the penultimate leaf blade was measured (Figure 1C). At the same time, the leaves of 9th–14th were peeled with tweezers, and the leaf area was measured by the ImageJ to the collected images.

### 4.3. Determination of Chlorophyll Content

After 6 weeks of growth in the hydroponic system, the 9th–14th leaves were collected, washed, and cut into small pieces. Chlorophyll (Chl) was extracted from about 0.1 g of leaves with 5 mL of 80% (*v*/*v*) acetone. The absorbance of the extracted chlorophyll a (Chl a) and chlorophyll b (Chl b) was measured at 645 nm and 663 nm, respectively. The levels of Chl a, Chl b, and total Chl (Ct) in the samples were calculated as previously described [86].

### 4.4. Phytohormone Analysis

After 6 weeks of growth under hydroponic conditions, all shoots of the SD and XJ salt cress ecotypes were collected, snap-frozen with liquid nitrogen, and stored at −80 °C. The concentrations of cytokinins (CKs) were determined by Enzyme-linked Immunosorbent Assay (Suzhou Keming Biotechnology Co., Ltd., Suzhou, China), and indole-3-acetic acid (IAA) was measured by Agilent 1100 high-performance liquid chromatograph and Kromasil C18 column (250mm × 4.6mm, 5 µm).

### 4.5. The Measurement of ACO Concentration

Similarly, whole shoots from 6-week-old SD ecotype and XJ ecotype seedlings were collected, and the concentration of 1-aminocyclopropane-1-carboxylate oxidase (ACO) was determined by Enzyme-Linked Immunosorbent Assay (Shanghai Enzyme-Linked Biotechnology Co., Ltd., Shanghai, China) using the RT-6100 Microplate Reader (Rayto, Shenzhen, China).

### 4.6. Protein Extraction and 2-DE Analysis

Total protein was extracted from plant tissue as described by Giavalisco [87], with minor modifications. Frozen plant tissue was ground to a fine powder under liquid nitrogen, and then proteins were extracted in ice-cold extraction solution [10% (*w*/*v*) tricarboxylic acid (TCA) in acetone with 0.07% (*v*/*v*) β-mercaptoethanol] for 1 h at −20 °C. Homogenates were centrifuged at 40,000× *g* for 30 min at 4 °C. After removing the supernatant, the precipitate was suspended in 100% acetone solution, cooled at −20 °C for 1 h and centrifuged at 40,000× *g* for 30 min at 4 °C. This process was repeated 2–3 times until the supernatant was colorless. The precipitate was then vacuum dried and stored at −80 °C or proceeded to the next step. The remaining pellets were dissolved in lysis buffer [7 M urea, 2.5 M Thiourea, 65 mM DTT, 4% (*w*/*v*) CHAPS] at room temperature for 1 h, and centrifuged at 40,000× *g* for 30 min at 4 °C. The supernatant was collected and stored at −80 °C. The protein concentration of each extract was determined by the Bradford method [88].

For each sample, 1.5 mg protein dissolved in lysis buffer was adjusted to a final volume of 450 µL with rehydration buffer containing 8 M urea, 15 mM DTT, 2% (*w*/*v*) CHAPS, 0.5% (*v*/*v*) IPG buffer (pH 4–7), and then it was loaded onto linear 24 cm dry IPG strips (pH 4–7, GE Healthcare Life Science, USA). These strips were then used for the IEF of the Ettan IPGphor II isoelectric focusing system using the following settings according to the manufacturer’s instructions (Amersham Biosciences, Uppsala, Sweden): 150 V for 1 h, 30 V for 6 h, 60 V for 6 h, 200 V for 1 h, 500 V for 1 h, 1000 V for 1 h, gradient to 8000 V for 1 h, and finally 8000 V for 7 h. Before separation in the second dimension, the strips were equilibrated twice for 15 min in equilibration buffer (50 mM Tris-HCl, pH 8.8, 6 M urea, 30% glycerol (*v*/*v*), 2% SDS (*w*/*v*), and 0.002% (*w*/*v*) bromophenol blue) with 1% DTT for first time, and for second time, 2.5% iodoacetamide instead of 1% DTT in the equilibration solution). The second dimensional SDS-PAGE was performed on a 1 mm thick 12.5% SDS-PAGE gel [89]. The 2-DE experiment was repeated 3 times using protein samples prepared from SD and XJ *Eutrema*, respectively. Proteins were visualized by Coomassie brilliant blue R250 staining, and gel images were taken using an image scanner (GE Healthcare, Chicago, IL, USA). Image analysis was performed using Image Master 2D Platinum software version 5.0 (Amersham Biosciences, Uppsala, Sweden). The experimental Mr (kDa) for each protein was estimated by comparison to protein markers, and the experimental pI was determined by its migration across the IPG strip. The frequency of each protein spot was estimated in percent by volume (% Vol). Only those proteins with significant and reproducible changes were considered as differentially expressed proteins.

### 4.7. Protein Digestion and Identification via ESI-MS/MS

Digestion and identification of protein were performed according to the method of Peng et al. [90] with some modifications. Selected protein spots were manually excised from each 2-DE gel, cut into 1 mm^3^ gel slices, placed in 1.5 mL centrifuge tubes, and destained with 50 mM ammonium bicarbonate and 50% (*v*/*v*) methyl alcohol for 5–6 times until the gel slices are transparent. After the destaining solution was completely discarded, acetonitrile (ACN) was added to shrink the gel pieces. The acetonitrile (ACN) was then removed, and the shrunken gel pieces were vacuum dried. The dried gel slices were swollen at an ice bath in 50 mM ammonium bicarbonate (pH 8.0–8.5) containing gel sequencing grade trypsin (10 µg/mL; Promega, Madison, WI, USA), then digested at 37 °C for 16–18 h. The digested peptides were extracted from the gel slices with 50% acetonitrile containing 5% formic acid. The pooled peptides were lyophilized and then resuspended in 0.1% formic acid (FA) to a final volume of 50 µL. The 50 µL peptide fragments were automatically injected into the strong cation exchange column of the multidimensional liquid chromatography system by the autosampler of the ProteomeX workstation (Thermo Finnigan, USA). Peptides were eluted from the SCX column (0.32 × 100 mm, Thermo Hypersil-Keystone BioBasic) by salt steps with increasing NH_4_Cl concentration (0, 50, 70, 100, 150, 400, 700, 1000 mM NH_4_Cl). These peptide fractions were collected and desalted onto two reverse-phase C18 columns (0.18 × 100 mm, Thermo Hypersil-Keystone BioBasic) and then treated with a gradient of acetonitrile solvent B (ACN in 0.1% FA) from 5 to 65% screened over 31 min, from 65 to 80% for 5 min, then hold 5 min at 80%, reset 5% for 1 min, rebalance 5% for 15 min. SCX and RP gradients were alternately synchronized over 140 min. The eluted peptides were directly loaded into an LCQ-Deca XP plus mass spectrometer (Thermo Electron, USA) for ESI-MS/MS detection. MS/MS data with default parameters were searched using the SEQUEST algorithm. The TurboSEQUEST program in Bioworks 3.0 software retrieved Arabidopsis data from the SWISS-PROT/TrEMBL proteome database. The identified peptides were further evaluated by charge state and cross-correlation number (Xcorr). Peptide matching criteria for cross-correlation scores were as follows: Xcorr > 1.5 for singly-charged ions, Xcorr > 2.0 for doubly-charged ions, and Xcorr > 2.5 for triply-charged ions and a correlation score (ΔCn) > 0.100. Only the best matching peptides are considered [91].

### 4.8. Quantitative Real Time PCR

Total RNA was extracted from two ecotypes of *Eutrema* plants by the method of Chomczynski and Sacchi [92]. In total, 2 µg of total RNA was reverse transcribed, and cDNA was generated using the FastQuant RT kit (with gDNase, TIANGEN). The collected cDNA was used as a template for quantitative real-time PCR (qRT-PCR). The *actin 2* gene served as a control to normalize target gene quantities [93]. The gene-specific qRT-PCR primers are listed in Appendix A. qRT-PCR was performed in a Light Cycler^®^ 96 thermal cycler Instrument (Roche Applied Science, Penzberg, Germany) using SYBR Green I (Roche): for the following reactions: 94 °C 30 s; 94 °C 5 s, 60 °C 30 s for 40 cycles and 95 °C 15 s, 60 °C 60 s, 95 °C 1 s. All reactions were replicated from three independent experiments. The formula for calculating the relative expression is as follows: ratio = 2^−ΔΔCt^ = 2 ^−(ΔCtt–ΔCtc)^ [94].

### 4.9. Functional Annotation

Protein spots identified by mass spectrometry were amplified in *A. thaliana* NCBI. The enriched GO terms of *Arabidopsis* homologues of these protein stains were detected using Goatools [95].

### 4.10. Statistical Analysis

All data obtained in this study were performed with at least three biological replicates. SPSS 17.0 software was used for statistical analysis and *t*-test and the significance level was 5%. Data are presented as mean ± standard deviation.

## 5. Conclusions

By comparing the proteome of shoots or roots of *Eutrema* SD and XJ ecotypes, this study identified several proteins that are closely related to defense and stress, aging, and energy metabolism. In China, *Eutrema* is distributed in Xinjiang along the Yellow River to the mouth of the Yellow River (Dongying, Shandong). Located in the Yellow River Delta, Shandong has a temperate monsoon climate, and its rainfall is 4–5 times that of Urumqi, Xinjiang. It can be seen that there are some proteins related to defense and stress resistance in the *Eutrema* SD ecotype. The relationship between defensive resistance and stress avoidance mechanisms is further explored. Xinjiang is a landlocked area with a temperate continental dry climate with little rainfall and seasonal concentrated precipitation. XJ *Eutrema* has evolved into a short life cycle xerophyte with a stress avoidance mechanism due to its high expression of senescence and metabolism-related proteins. Proteomic studies of *Eutrema* will help us to understand the specificity of extremophile physiological and metabolic devices, summarize the evolutionary process of halophytes, and provide a valuable resource for further deciphering the genetic mechanisms of local adaptation in this model plant.

## Figures and Tables

**Figure 1 ijms-23-06124-f001:**
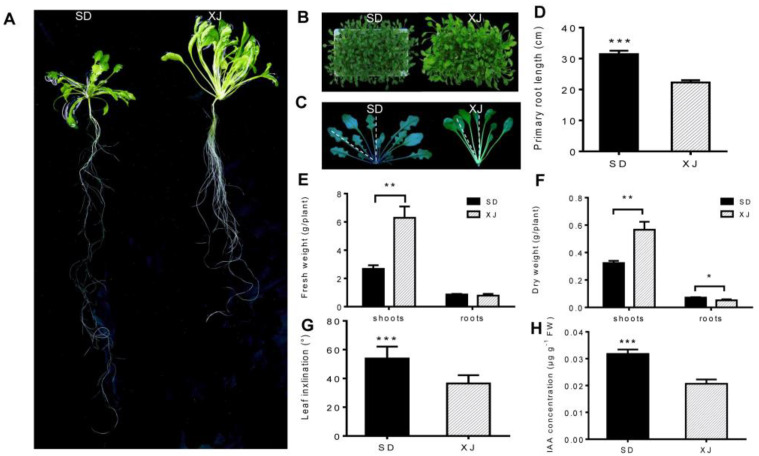
Growth and morphological analysis of SD and XJ plants cultured in a hydroponic system for 6 weeks. (**A**,**B**) Growth status of SD and XJ plants. (**C**) Comparison of leaf inclination angles of SD and XJ plants after 6 weeks of hydroponics. (**D**–**F**) The primary root length, fresh weight (FW), and dry weight (DW) of two *Eutrema* were examined. (**G**) Leaf inclination angle measurements were made between the central axis and the penultimate leaf of two *Eutrema*. (**H**) Concentration of IAA in shoots. Data were expressed as means ± SD and *t*-test was used for statistical analysis. (* *p* < 0.05, ** *p* < 0.01 and *** *p* < 0.001). All experiments were tripled, and each experiment contained at least 9 plants.

**Figure 2 ijms-23-06124-f002:**
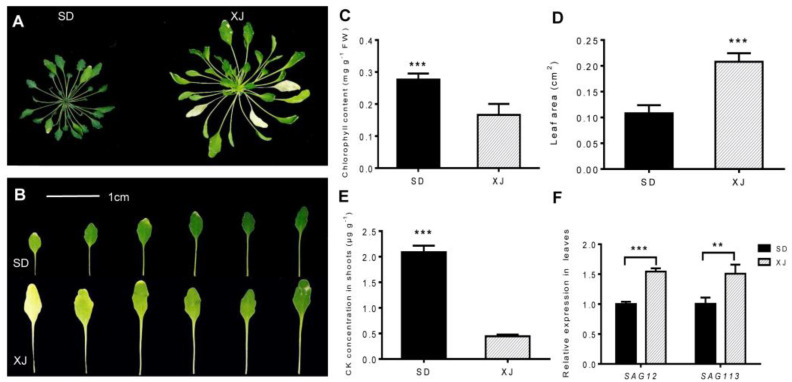
Senescence analysis of SD and XJ plants cultured in a hydroponic system for 6 weeks. (**A**) The growth state of the shoots. (**B**) Comparison of SD and XJ plants with 9th–14th leaves. Bars = 1 cm. (**C**) Total chlorophyll content in leaves of SD and XJ plants. (**D**) Leaf area values for SD and XJ plants. (**E**) The concentration of CK in SD and XJ shoots. Concentrations of CK in SD and XJ. (**F**) Expression of *SAG12* and *SAG113* in leaves of two ecotypes. Data are presented as means ± SD and *t*-test was used for statistical analysis. (** *p* < 0.01 and *** *p* < 0.001).

**Figure 3 ijms-23-06124-f003:**
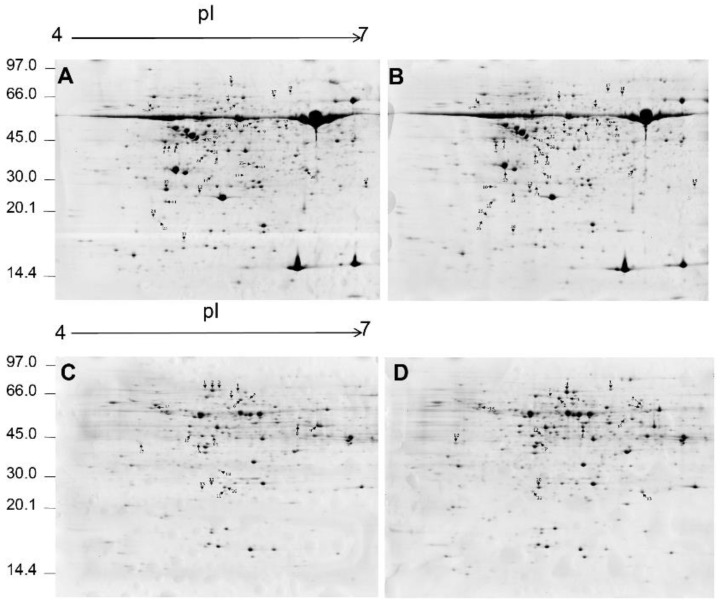
The 2-DE profiles of SD and XJ *Eutrema* proteins under normal hydroponic conditions for 6 weeks. The marked spots were identified by ESI-MS/MS. (**A**) 2-DE protein gel was from SD *Eutrema* shoots. (**B**) 2-DE protein gel was from XJ *Eutrema* shoots. (**C**) 2-DE protein gel extracted from SD *Eutrema* root. (**D**) 2-DE protein gel extracted from XJ *Eutrema* root.

**Figure 4 ijms-23-06124-f004:**
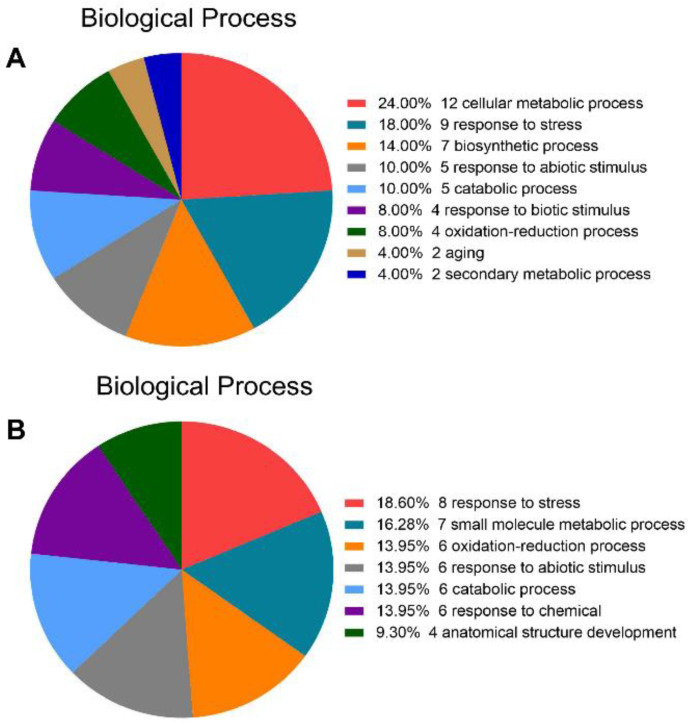
Functional annotation of different proteins under normal hydroponic conditions. (**A**) Biological processes of proteins identified in shoots. (**B**) Biological processes of proteins identified in roots.

**Figure 5 ijms-23-06124-f005:**
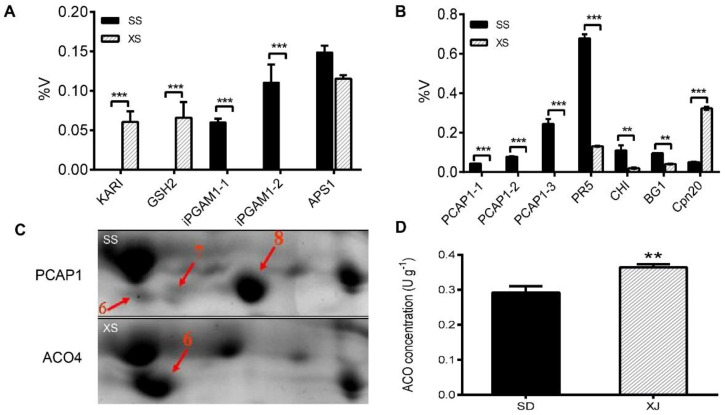
Expression patterns of representative differentially expressed proteins in shoots from two *Eutrema* ecotypes under normal hydroponic conditions. (**A**) Quantitative analysis of differentially expressed protein (DEPs) species spots related to cellular metabolic processes. (**B**) Quantification of DEPs spots related to stress responses. (**C**) Changes in the expression of the ACO4 protein spot in the shoot. (**D**) ACO concentrations in SD and XJ shoots. Statistical analysis was performed on the normalized volume percentage (% Vol) of protein spots in 3 replicate biological samples using the mean ± SD method and statistical calculations were performed using the t-test. (** *p* < 0.01 and *** *p* < 0.001). The abbreviations for Figure 5 were: SS, shoot of SD ecotype; XS, shoot of XJ ecotype; KARI, ketoacid reductase isomerase; GSH2, glutathione synthetase; iPGAM1, 2,3-bisphosphoglycerate-independent phosphoglycerate mutase 1; iPGAM1−1, spot SS4; iPGAM1−2, spot SS5; APS1, ATP sulfurylase 1; PCAP1, plasma membrane-associated cation-binding protein 1; PCAP1−1, spot SS6; PCAP1−2, spot SS7; PCAP1−3, spot SS8; PR5, pathogenesis-related protein 5; CHI, endochitinase; BG1, glucan endo−1, 3-beta-glucosidase; Cpn 20, 20 kDa chaperonin; ACO4, 1-aminocyclopropane-1-carboxylate oxidase 4.

**Figure 6 ijms-23-06124-f006:**
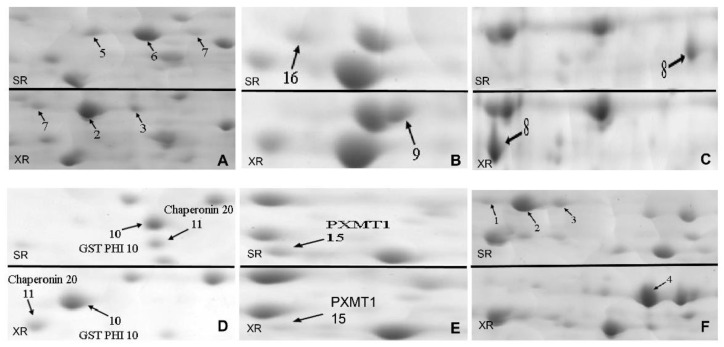
Expression patterns of some root protein spots on the 2-DE map. Marked points are differentially expressed proteins. (**A**) 2,3-bisphosphoglycerate-independent phosphoglycerate mutase 1 (iPGAM1) (spots SR5, 6 and 7, spots XR7, 2 and 3). (**B**) pyruvate dehydrogenase E1 component subunit beta-1 (MAB1) (spots SR16, XR9). (**C**) monodehydroascorbate reductase 2 (MDAR2) (spots SR8, XR8). (**D**) glutathione S-transferase F10 (GST PHI10) (spots SR10, XR10) and 20 kDa chaperonin (Chaperonin 20, Cpn20) (spots SR11, XR11). (**E**) paraxanthine methyltransferase 1 (PXMT1) (spots SR15, XR15) and (**F**) jacalin-related lectin 34 protein (JRL34) (spots SR1, 2, 3, and spots XR4). The abbreviations for Figure 6 were: SR, root of SD ecotype; XR, root of XJ ecotype.

**Figure 7 ijms-23-06124-f007:**
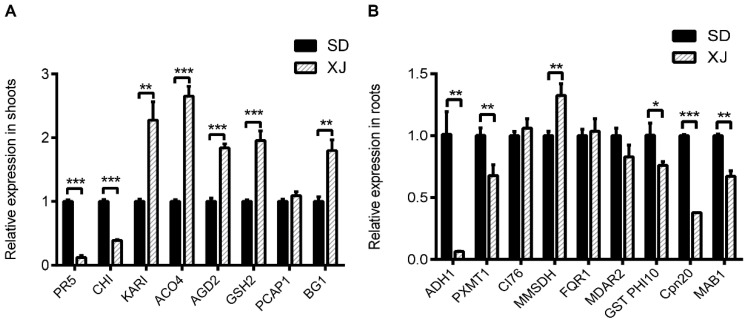
Relative transcript levels in two *Eutrema* plants under normal hydroponic conditions for 6 weeks. (**A**) Differentially expressed genes in shoots. (**B**) Differentially expressed genes in roots. Data are presented as mean ± SD obtained from 3 biological replicates. *t*-test was used to analyze the changes in the gene expression (* *p* < 0.05, ** *p* < 0.01, and *** *p* < 0.001). The abbreviations for Figure 7 were: PR5, pathogenesis-related protein 5; CHI, endochitinase; ACO4, 1-aminocyclopropane-1-carboxylate oxidase 4; AGD2, LL-diaminopimelate aminotransferase; GSH2, glutathione synthetase; PCAP1, plasma membrane-associated cation-binding protein 1; BG1, glucan endo-1,3-beta-glucosidase; KARI, ketoacid reductase isomerase; ADH1, alcohol dehydrogenase class-P; PXMT1, paraxanthine methyltransferase 1; CI76, NADH dehydrogenase [ubiquinone] iron-sulfur protein 1; MMSDH, methylmalonate-semialdehyde dehydrogenase; FQR1, NAD(P)H dehydrogenase (quinone); MDAR2, monodehydroascorbate reductase 2; GST PHI10, glutathione S-transferase F10; Cpn20, 20 kDa chaperonin; MAB1, pyruvate dehydrogenase E1 component subunit beta-1.

**Table 1 ijms-23-06124-t001:** Differentially expressed proteins identified in shoots of SD and XJ ecotype of *Eutrema*.

Spot		NCBI Accession		Exper. ^e^	Theor. ^f^			
No. ^a^	Locus. ^b^	No. ^c^	Protein Name. ^d^	pI/Mr	pI/Mr	Score. ^g^	NP. ^h^	Pattern. ^i^
	Cellular metabolic process							
XS4	At3g58610	XP_006402768	ketol-acid reductoisomerase	5.8/56	5.64/57.3	20	2	appear
XS19	At5g27380	XP_024007192	glutathione synthetase	5.8/56	5.52/53.9	20	2	appear
XS8	At4g33680	XP_024005372	LL-diaminopimelate aminotransferase	5.6/44	6.38/46.5	20	2	appear
SS4	At1g09780	XP_006417539	2,3-bisphosphoglycerate-independent phosphoglycerate mutase 1	5.45/59	5.32/60.5	78	7	decrease
SS5	At1g09780	XP_006417539	2,3-bisphosphoglycerate-independent phosphoglycerate mutase 1	5.6/59	5.32/60.5	384	36	appear
SS27	At3g22890	XP_006406118	ATP sulfurylase 1	6/44	6.34/51.5	70	7	increase
XS28	At3g22890	XP_006406118	ATP sulfurylase 1	6/44	6.34/51.5	60	6	decrease
SS21	At1g12050	XP_006417270	fumarylacetoacetase	5.6/44	5.23/46.1	40	4	appear
XS33	At2g40010	XP_006411245	60S acidic ribosomal protein P0-1	5.3/34	5.19/33.7	100	9	equal
	Response to stress							
SS6	At4g20260	XP_006413890	plasma membrane-associated cation-binding protein 1	5/36	4.65/24.7	98	8	appear
SS7	At4g20260	XP_006413890	plasma membrane-associated cation-binding protein 1	5/36	4.65/24.7	66	5	appear
SS8	At4g20260	XP_006413890	plasma membrane-associated cation-binding protein 1	5/37	4.65/24.7	134	9	appear
SS10	At1g75040	XP_006390358	pathogenesis-related protein 5	5/25	4.65/22.8	200	20	increase
SS14	At2g43570	XP_006397529	endochitinase CHI	5.3/27	5.84/29.8	40	4	increase
SS29	At3g57270	XP_006409034	probable glucan endo-1,3-beta-glucosidase BG1	5.3/35	8.94/37.7	20	2	increase
XS15	At2g30860	XP_024004280	glutathione S-transferase F9	6.7/25	6.17/24.1	30	3	decrease
	Response to abiotic stimulus							
XS12	At5g20720	XP_024011493	20 kDa chaperonin	5.3/25	5.23/21.4	170	16	increase
SS11	At3g11630	XP_006407428	2-Cys peroxiredoxin BAS1	5/23	5.01/22.4	460	46	increase
SS12	At3g11630	XP_006407428	2-Cys peroxiredoxin BAS1	5.3/24	5.01/22.4	30	3	decrease
XS11	At3g11630	XP_006407428	2-Cys peroxiredoxin BAS1	5/24	5.01/22.4	88	9	decrease
Aging								
XS6	At1g05010	XP_006418070	1-aminocyclopropane-1-carboxylate oxidase 4	5/37	5.24/36.7	60	4	appear
Unknown								
SS26	At3g01500	XP_006408514	beta carbonic anhydrase 1	6.3/33	6.14/25.6	80	7	equal
XS27	At3g01500	XP_006408514	beta carbonic anhydrase 1	6.3/26.3	6.14/25.6	66	2	equal
SS30	At1g55480	XP_006392635	protein MET1	5.3/34	8.36/37.4	30	3	equal
XS32	At1g55480	XP_006392635	protein MET1	5.3/35	8.19/37.4	256	23	equal

^a^ Assigned spot number as indicated in Figure 3. SS refers to the shoots of SD ecotype of *Eutrema*. XS refers to the shoots of XJ ecotype of *Eutrema*. SR refers to the roots of SD ecotype of *Eutrema*. XR refers to the roots of XJ ecotype of *Eutrema*. ^b^ Gene symbol in *Arabidopsis*. ^c^ Database accession numbers of *Eutrema* from the NCBI database. ^d^ The identified proteins name in *Eutrema*. ^e^ Experiment mass (kDa) and pI of identified proteins. ^f^ Theoretical mass (kDa) and pI of identified proteins. ^g^ The sequest score. ^h^ Number of peptides sequenced. ^i^ Pattern of protein spots on 2D gel. “Appear” means that this spot appears in one ecotype, and there is no corresponding spot in the other ecotype. “Increase” means that the optical density of the corresponding spot in the ecotype is greater than or equal to 2 times (*p* < 0.05). “Decrease” indicates that the optical density of the corresponding spot in the ecotype is less than 1/2 (*p* < 0.05). “Acid” means that the spot in one ecotype is more acidic than its corresponding spot in another ecotype. “Basic” means that the spot in one ecotype is more basic than the spot in another ecotype.

**Table 2 ijms-23-06124-t002:** Differentially expressed proteins identified in roots of SD and XJ ecotype of *Eutrema*.

Spot		NCBI Accession		Exper. ^e^	Theor. ^f^			
No. ^a^	Locus. ^b^	No. ^c^	Protein Name. ^d^	pI/Mr	pI/Mr	Score. ^g^	NP. ^h^	Pattern. ^i^
	Small molecule metabolic process							
SR5	At1g09780	XP_006417539	2,3-bisphosphoglycerate-independent phosphoglycerate mutase 1	5.68/65	5.32/60.6	68	7	basic
SR6	At1g09780	XP_006417539	2,3-bisphosphoglycerate-independent phosphoglycerate mutase 1	5.72/65	5.32/60.6	490	42	basic
SR7	At1g09780	XP_006417539	2,3-bisphosphoglycerate-independent phosphoglycerate mutase 1	6/65	5.32/60.6	80	7	basic
XR2	At1g09780	XP_006417539	2,3-bisphosphoglycerate-independent phosphoglycerate mutase 1	5.5/65	5.27/60.5	128	12	acid
XR3	At1g09780	XP_006417539	2,3-bisphosphoglycerate-independent phosphoglycerate mutase 1	5.6/65	5.27/60.5	154	14	acid
XR7	At1g09780	XP_006417539	2,3-bisphosphoglycerate-independent phosphoglycerate mutase 1	5.4/65	5.27/60.5	80	6	acid
XR5	At5g08570	XP_006399365	pyruvate kinase	6.3/64	5.93/55	70	6	appear
SR16	At5g50850	XP_006402055	pyruvate dehydrogenase E1 component subunit beta-1	5.1/38	5.11/35.9	32	4	decrease acid
XR9	At5g50850	XP_006402055	pyruvate dehydrogenase E1 component subunit beta-1	5.3/36	5.11/35.9	58	5	increase basic
SR4	At2g45290	XP_024010889	transketolase-2	5.7/67	5.64/68.9	50	4	appear
	Oxidation-reduction process							
XR1	At5g37510	XP_006405867	NADH dehydrogenase [ubiquinone] iron-sulfur protein 1	6/68	5.72/77.9	70	6	appear
XR13	At5g54500	XP_006401566	NAD(P)H dehydrogenase (quinone) FQR1	6.3/25	5.96/21.8	136	11	increase
SR8	At5g03630	XP_024011589	monodehydroascorbate reductase 2	6.3/45	5.25/47.5	166	12	decrease basic
XR8	At5g03630	XP_024011589	monodehydroascorbate reductase 2	5.7/43	5.25/47.5	108	9	increase acid
XR6	At2g14170	XP_006409665	methylmalonate-semialdehyde dehydrogenase [acylating]	6.3/64	8.97/64.7	82	7	appear
	Response to abiotic stimulus							
SR9	At1g77120	XP_006390122	alcohol dehydrogenase class-P	6.5/45	5.83/41.2	120	12	increase
SR10	At2g30870	XP_006410185	glutathione S-transferase F10	5.5/25	5.49/24.1	50.11	5	decrease basic
XR10	At2g30870	XP_006410185	glutathione S-transferase F10	5.3/25	5.49/24.1	148	15	increase acid
SR11	At5g20720	XP_024011493	20 kDa chaperonin	5.5/25	5.23/21.4	284	26	decrease basic
XR11	At5g20720	XP_024011493	20 kDa chaperonin	5.2/25	5.23/21.4	262	21	increase acid
SR15	At1g66700	XP_006391421	paraxanthine methyltransferase 1	5/42	5.34/39.8	68	6	increase
SR1	At3g16460	XP_006406833	jacalin-related lectin 34	5.3/68	5.31/72.5	50	5	acid
SR2	At3g16460	XP_006406833	jacalin-related lectin 34	5.4/68	5.31/72.5	148	14	acid
SR3	At3g16460	XP_006406833	jacalin-related lectin 34	5.5/68	5.31/72.5	40	4	acid
XR4	At3g16460	XP_006406833	jacalin-related lectin 34	5.6/67	5.31/72.5	240	23	basic
	Unknown							
SR18	At1g48090	XP_006393484	uncharacterized LOC18010485	5.3/26	6.33/46.1	42	4	appear
XR14	At1g28680	XP_006415631	spermidine sinapoyl-CoA acyltransferase	6.1/56	5.65/49.7	34	3	appear
XR16	At5g43060	XP_006403303	probable cysteine protease RD21B	4.6/50	5.89/32.5	140	13	equal

^a^ Assigned spot number as indicated in Figure 3. SS refers to the shoots of SD ecotype of *Eutrema*. XS refers to the shoots of XJ ecotype of *Eutrema*. SR refers to the roots of SD ecotype of *Eutrema*. XR refers to the roots of XJ ecotype of *Eutrema*. ^b^ Gene symbol in *Arabidopsis*. ^c^ Database accession numbers of *Eutrema* from the NCBI database. ^d^ The identified proteins name in *Eutrema*. ^e^ Experiment mass (kDa) and pI of identified proteins. ^f^ Theoretical mass (kDa) and pI of identified proteins. ^g^ The sequest score. ^h^ Number of peptides sequenced. ^i^ Pattern of protein spots on 2D gel. “Appear” means that this spot appears in one ecotype, and there is no corresponding spot in the other ecotype. “Increase” means that the optical density of the corresponding spot in the ecotype is greater than or equal to 2 times (*p* < 0.05). “Decrease” indicates that the optical density of the corresponding spot in the ecotype is less than 1/2 (*p* < 0.05). “Acid” means that the spot in one ecotype is more acidic than its corresponding spot in another ecotype. “Basic” means that the spot in one ecotype is more basic than the spot in another ecotype.

## Data Availability

The raw mass spectrometry data and SEQUEST search results list corresponding to proteomic analysis were deposited in Proteome Xchange Consortium via the PRIDE partner repository with the data set identifier ‘PXD018751’.

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
