# Peer review of "Dissecting the Molecular Regulation of Natural Variation in Growth and Senescence of Two *Eutrema salsugineum* Ecotypes"

_ijms, 2022, doi:10.3390/ijms23116124_

Round 1
Reviewer 1 Report
Salt cress is a typical halopphyte and contains different ecotypes. In china, the salt cress from shangdong (SD) and xinjiang(XJ) are two typical ecotypes. In this paper, the authors compared the morphological and Physiological differences, and identified the differently expressed protein spots from these two ecotypes by 2-DE and ESI-MS/MS using plants under normal conditions to reveal the mechanism of selection of complex traits under natural variation. The results are interesting and provide some novel findings between different ecotypes. However, the descriptions of some paragraphs need to be improved before the publication in journal.
Please see below for my comments and suggestions that will help to improve the manuscript.
- Whether the results can support the mechanism of selection of complex traits? Therefore, whether the title “Dissecting the molecular regulation of natural variation in physiological signature of two Eutrema salsugineum ecotypes” is suitable?
- In result 2.1 , the authors detected the CK content, as I know, ABA and ethylene are important in leaf senescence, why not measure the ABA content?
- In result 2.3, many descriptions should move to the discussion section.
- The title need to be changed according to the results.
Author Response
Dear Professor,
We sincerely thank you for your affirmation and encouragement of our article. As you mentioned, our results of “Dissecting the molecular regulation of natural variation in physiological signature of two Eutrema salsugineum ecotypes” are interesting and provide some novel findings between different ecotypes. And we also give your many thanks for thoroughly examining our manuscript and providing very helpful comments on how to improve our manuscript. We have tried our best to revise the manuscript according to your kind suggestions. The detailed changes are given below.
- Based on the research results in our article, we changed the title to “Dissecting the molecular regulation of natural variation in growth and senscence of two Eutrema salsugineum ecotypes” to better fit the results of the article.
- Leaf senescence is a highly programmed developmental process regulated by phytohormones and environmental factors. Phytohormones, such as ethylene (ET), abscisic acid, jasmonic acid, and salicylic acid (SA), promote leaf senescence. Zuzana Kuˇcerová et al. (2020) has been shown that with the occurrence of chlorophyll decomposition, chloroplast degradation, and photosynthesis inhibition, leaf senescence can be inhibited by the application of exogenous cytokinins. Therefore, cytokinin concentrations maybe can explain the difference of two Eutrema salsugineum ecotypes.
- In discussion section, we focused on the part of “stress and defense-related proteins in shoots” and “proteins related to energy metabolism in roots”. Carefully considerate, we think this description may be an integral part.
- We revised the title based on the results of the article.
We sincerely hope that this revised manuscript can meet your requirements.
Once again, thank you for your good suggestions and comments.
We look forward to hearing from you.
Best regards,
Zenglan Wang
Reviewer 2 Report
The MS entitled "Dissecting the molecular regulation of natural variation in physiological signature of two Eutrema salsugineum ecotypes" with authors: Fanhua Wang, Zhibin Sun, Min Zhu, Qikun Zhang, Yufei Sun, Wei Sun, Chunxia Wu, Tongtong Li, Yiwu Zhao, Changle Ma, Hui Zhang, Yanxiu Zhao, Zenglan Wang shows an interesting study of two salt cress ecotypes. Authors compare the differences between two ecotypes grown under control conditions, using different physiological, biochemical and molecular (protein identification via 2 DE separation and ESI MS/MS identification; expression analyses via qRT-PCR). The data was interpreted in connection with the evolutionary differences at the protein level between both ecotypes .
It is necessary to improve figure subtitles because it is essential to state the conditions and age of plants for each figure. I suggest removing the methodological part from the figure subtitles.
In addition, it should be clear if all analyses were performed under normal (control) conditions or not. The growth stage (days/weeks after germination) also should be clearly stated for each method.
The necessary corrections are highlighted in the pdf version.
I suggest Minor revision.

Author Response
Dear Professor,
We sincerely thank you for generalization and affirmation of our article, and so many suggestions and comments, which are of great help to improve our manuscript. Now we have made the following revision.
- We improved figure subtitles and the experimental conditions are described below each figure. We have removed the methodological part from the figure subtitles.
- We supplemented the experimental conditions and the growth stage for each method.
- The necessary correction details are in the word version.
We are appreciated for your suggestions and hope that the correction can meet your requirements.
Once again, thank you for your good suggestions and comments.
We look forward to hearing from you.
Best regards,
Zenglan Wang